# Dynamic Emotion Recognition and Social Inference Ability in Traumatic Brain Injury: An Eye-Tracking Comparison Study

**DOI:** 10.3390/bs13100816

**Published:** 2023-10-03

**Authors:** Leanne Greene, John Reidy, Nick Morton, Alistair Atherton, Lynne A. Barker

**Affiliations:** 1Centre for Behavioural Science and Applied Psychology, Department of Psychology, Sociology and Politics, Sheffield Hallam University, Sheffield S10 2BP, UK; j.g.reidy@shu.ac.uk (J.R.); l.barker@shu.ac.uk (L.A.B.); 2Neuro Rehabilitation Outreach Team, Rotherham, Doncaster and South Humber NHS Trust, Doncaster DN4 8QN, UK; n.morton@nhs.net; 3Consultant Clinical Neuropsychologist, Atherton Neuropsychology Consultancy Ltd. Parkhead Consultancy, 356 Ecclesall Road, Sheffield S11 9PU, UK; aatherton@neuropsychol.net

**Keywords:** traumatic brain injury, social cognition, emotion recognition, eye tracking, fixation, visual processing, dynamic stimuli

## Abstract

Emotion recognition and social inference impairments are well-documented features of post-traumatic brain injury (TBI), yet the mechanisms underpinning these are not fully understood. We examined dynamic emotion recognition, social inference abilities, and eye fixation patterns between adults with and without TBI. Eighteen individuals with TBI and 18 matched non-TBI participants were recruited and underwent all three components of The Assessment of Social Inference Test (TASIT). The TBI group were less accurate in identifying emotions compared to the non-TBI group. Individuals with TBI also scored lower when distinguishing sincere and sarcastic conversations, but scored similarly to those without TBI during lie vignettes. Finally, those with TBI also had difficulty understanding the actor’s intentions, feelings, and beliefs compared to participants without TBI. No group differences were found for eye fixation patterns, and there were no associations between fixations and behavioural accuracy scores. This conflicts with previous studies, and might be related to an important distinction between static and dynamic stimuli. Visual strategies appeared goal- and stimulus-driven, with attention being distributed to the most diagnostic area of the face for each emotion. These findings suggest that low-level visual deficits may not be modulating emotion recognition and social inference disturbances post-TBI.

## 1. Introduction

Traumatic brain injury (TBI) is a global public health concern with approximately 69 million new cases each year [1]. The long-term sequelae of TBI include physical (i.e., headaches), cognitive (i.e., executive function, concentration), affective (i.e., mood disorders), and social cognition (i.e., diminished theory of mind) impairments [2]. Importantly, social cognition impairments are documented across all severities of TBI [3,4], and diminished abilities are associated with poor patient outcomes [5,6,7], and negatively impact long-term family functioning and wellbeing [8]. Despite the likelihood that social cognition impairments after TBI are underreported [9,10], the evidence base for their prevalence is strong [11,12,13,14]. One prominent area of disruption post-TBI is facial emotion recognition [11,14,15,16], with estimates that up to 39% of individuals with moderate-to-severe TBI may exhibit this impairment [17]. This is highly problematic, as facial expressions are one of the richest tools for social inference and communication [18]. Additionally, individuals with TBI display difficulties in interpreting social cues such as tone of voice and gestures that impede the understanding of other people’s feelings, thoughts, and intentions [14,19,20].

Although emotion recognition impairments are well documented post-TBI, the mechanisms underpinning these changes are not fully understood. Specifically, it is unknown whether they stem from disruption to early visual processing (i.e., aberrant eye fixation patterns) or social cognition, or a combination of both. An association between eye movement dysfunction and poor emotion recognition has been documented for neurological disease and injury [21,22,23,24,25,26]. Although research on TBI populations is scant, it is unsurprising that eye fixation patterns are disrupted, as anatomically, several frontal brain areas modulating eye movements are susceptible to pathology due to their position within the skull and the trajectory of the brain on impact [27,28]. Furthermore, emotion and social inference recognition are complex processes, requiring several synchronised visual strategies (e.g., saccades, fixations) of the body and face. This complexity is increased as different emotional expressions elicit distinct fixation scan path patterns; for example, angry and sad faces produce earlier and longer fixations on the eye region, compared to happy faces wherein the mouth receives more attention [29,30,31,32]. Unfortunately, the emotion recognition field has been hampered by an experimental bias towards behavioural data, with a paucity of studies exploring visual strategies of real-time physiological data. Furthermore, there has been a growing discussion regarding the poor ecological validity of frequently implemented static facial expression stimuli, and a call for increased use of dynamic stimuli [11,33]. The current study had three aims:(1)To explore dynamic emotion recognition and social inference abilities between adults with and without TBI;(2)To determine if adults with TBI exhibited different fixation patterns compared to adults without TBI in response to dynamic social interactions;(3)To investigate the relationships between fixation patterns, emotion recognition, and social inference accuracy scores.

## 2. Materials and Methods

### 2.1. Participants

The study included 18 TBI participants and 18 matched non-TBI participants (N = 36). Due to a dataset error, there was one less non-TBI participant for the social inference minimal (SI-M) and social inference enhanced (SI-E) components (N = 35). Ethical approval was obtained through Leeds East NHS Research Ethics Committee and Sheffield Hallam University Faculty Research Ethics. TBI participants were recruited through two brain injury rehabilitation services within the UK National Health Service. The non-TBI participants were recruited via stratified opportunity and snowballing sampling, and matched to the TBI participants for sex, age, and education. All participants provided informed consent and completed all three components of The Assessment of Social Inference Test (TASIT) [14] whilst undergoing eye tracking.

TBI information was acquired from medical records, including imaging scan data and hospital admission notes. Participant pathology was heterogeneous, including brain haemorrhage, skull fracture, and contusion, which is typical for this patient cohort [22]. The mechanisms of injury included assault, road traffic accidents, falls, and pedestrian collisions. Thirteen participants had frontal lobe pathology, and five had pathology outside of frontal cortices to other cortical and/or subcortical regions (see Appendix A). As anticipated, we recruited more males than females, which is in line with existing evidence that males are more likely to sustain a brain injury compared to females [23,24]. Years of education did not differ between groups, but the group with TBI fell significantly below the non-TBI group for all three components of the IQ test. This is a pattern frequently reported in the brain injury literature [18,19]. The non-TBI group all reported as neurotypical with no history of brain injury or concussion. See Table 1 for the demographic characteristics of the TBI participants and non-TBI groups.

#### 2.1.1. Inclusion/Exclusion Criteria

TBI inclusion criteria

TBI sustained in adulthood is typically determined by assessment at hospital admission or brain pathology based on imaging scans.

Participants were required to be at least one-year post injury to ensure that the chronic rather than acute effects of brain injury were measured.Patients were aged between 18 and 65 to account for any effects of natural aging.

Exclusion criteria for all participants

History of psychiatric illness;Severe recent drug and alcohol abuse assessed by Michigan Alcohol Screening Test (MAST) [25] and Drug Abuse Screening Test (DAST) [26];Significant depression and anxiety, found using the Hospital Anxiety and Depression Scale (HADS) [27];Visual deficits using the Visual Object and Space Perception Battery (VOSPB) [28] Warrington and James, 1991), and Cortical Vision Screening Test (CORVIST) [29].

Collaborating referring clinicians did not document how many participants were excluded based on these criteria. Nonetheless, among the individuals with TBI who were screened as eligible, the group did exhibit a notably elevated composite score on the HADS compared to individuals with no TBI (anxiety M = 8.11, SD = 4.61, depression M = 6.00, SD = 3.31, overall HADS M = 14.11, SD = 6.28); matched non-TBI-group (anxiety M = 4.94, SD = 2.04, depression M = 2.67, SD = 1.94, overall HADS M = 7.61, SD = 3.48), but this pattern is not unusual in post-TBI populations, even if the injury is mild [30,31].

#### 2.1.2. Injury Severity

Currently, there are two primary brain injury severity measurements, post-traumatic amnesia (PTA) and loss of consciousness (LOC) based on the Glasgow Coma Scale (GCS) [32]. PTA can be defined as the period between a TBI occurring and the patient regaining full awareness and standard memory function [33]. PTA less than 24 h is indicative of mild TBI, greater than 24 h but 7 days or less is moderate, and PTA longer than 7 days indicates severe injury. This study cohort included 16 severe, 1 moderate, and 1 mild TBI. The GCS assesses motor, verbal, and eye responses, providing a score between 3 and 15. Scores between 3 and 8 indicate severe injury, between 9 and 12 moderate, and between 13 and 15 mild injuries. Collaborating referring clinicians determined the severity of injury rating based on the standard procedures of LOC and PTA.

### 2.2. Design

A quasi-experimental design was utilised to compare the social cognition abilities of participants with and without TBI. A Tobii T120 Eye Tracker [34] recorded participants’ eye movements, and Tobii software was used to calculate metrics including (1) time to first fixation, (2) first fixation duration, (3) total fixation duration, and (4) total fixation count. Three areas of interest (AOI) were chosen for the study: the eyes, nose, and mouth of the actors in the TASIT task.

#### 2.2.1. Stimuli and Procedure

The TASIT [14] comprises video vignettes of professional actors engaged in social interactions. The actors were employed to use a ‘method style’ of acting, wherein they were required to demonstrate an emotion relevant to the situation. This method provided a realistic, spontaneous, and natural test context. The TASIT includes three sub-component measures: the emotion evaluation test (EET), the social inference-minimal test (SI-M), and the social inference-enriched (SI-E) test. All three sub-components of the TASIT were administered during the study. In brief, the EET assessed understanding of communication components such as facial expressions, tone of voice, and gestures. The SI-M provided a measure of the participant’s ability to read paralinguistic features, facial expressions, and tone of voice in either sincere or sarcastic situations. Finally, the SI-E assessed the participant’s ability to use contextual cues (i.e., tone of voice and facial expressions) to determine if everyday conversations were deceptive or sarcastic. The primary behavioural score for each component was the accuracy score (i.e., identifying the correct emotion or expression). A more detailed description of the TASIT sub-components can be found in Appendix B. Only correct responses were included in all analyses. There was one practice trial at the beginning of each sub-component, and participants verbally indicated their response, which the researcher documented. Appendix C outlines the experimental process in more detail. It took approximately 90 min to administer the TASIT.

Note: Two participants with TBI opted out of the Wechsler Abbreviated Scale of Intelligence [34] (IQ test). Due to a technical issue, one participant’s data were corrupted for the non-TBI group SI-M and SI-E, but this made little difference to the demographic comparisons. A Mann–Whitney U test was used to compare group differences, and an asterisk (*) indicates significant scores.

The TASIT is a superior social inference task, as it includes naturally occurring visual cues during social situations (the demeanour of the speaker and reaction of the listener), which many other experimental inference tasks do not. For example, while the Movie for the Assessment of Social Cognition (MASC) [35] includes dynamic social cues, it does not explore the processing of sincere and sarcastic exchanges in as much depth as the TASIT. It has been reported that the TASIT has good reliability and high test–retest reliability levels (*r* = 0.74–0.88), as well as good construct validity, low practice effects, and high ecological validity [14,36]. The TASIT is gaining traction in brain injury research [37,38,39].

#### 2.2.2. Apparatus Eye-Tracker

A Tobii T120 eye-tracker and Tobii Studio Eye Tracking Analysis software (Tobii Studio version 3) [40], connected to a Windows-based PC running Windows version 7, were used to collect and analyse eye-tracking data. The eye-tracker had a 17-inch thin-film-transistor screen (1280 × 1024 pixels) with an embedded infrared camera. The technology does not have a head frame, ensuring participants are free to exhibit natural head and eye movements. Eye tracking data were sampled at 120 Hz with an accuracy of 0.5°. The default Tobii fixation filter algorithm was used for all three components of the TASIT (fixation threshold at 35 pixels for velocity and 35 pixels for distance per sample). An eye movement velocity above the 0.5° per second threshold was classified as a saccade sample, and below it was classed as a fixation. The standard five-point calibration of each eye was undertaken for each component of the TASIT for each participant. Participants were seated approximately 62 cm away from the screen in a stationary chair.

#### 2.2.3. Eye Tracking Metrics

Eye tracking metrics were calculated using Tobii software [40]. The metrics analysed were ‘total fixation duration (seconds)’, which was the sum of all fixations within an active area of interest (AOI) (e.g., how long the participant spent looking at the eyes, the nose and the mouth), and ‘fixation count (count)’, which indicates the number of times the participant fixates on an active AOI. Fixation counts are amalgamated throughout the experiment as participants redirect their fixations back and forth between AOIs.

## 3. Results

Statistical Package for Social Sciences (SPSS) version 23 was used to analyse data [41]. Parametric assumptions were checked for raw data. The behavioural data were relatively normally distributed, but the eye-tracking data had moderate violations of normality regarding skewness and kurtosis, homogeneity of variance, and outliers. As violations were moderate and not severe, and because the study had approximately equal sample sizes, the effects of violation were thought minimal [42]. Furthermore, to guard against the correction of accurate but ‘non-normal’ data, and to avoid the challenges associated with transforming skewed data, parametric analyses were conducted on untransformed data [43]. Although the present study recruited similar participant numbers to previous dynamic social inference research in different populations [44,45], the sample size is still modest and, as such, the α for main inferential analyses was set at 0.05, with adjusted α = 0.01 for all post hoc follow-up analyses (α = 0.01).

### 3.1. Emotion Evaluation Test

For the EET, a 2 (TBI vs. no TBI group) * (7—emotion) ANOVA was conducted to explore potential differences in emotion perception. A further 2 * (7) * (3—eyes, nose, mouth AOI) repeated-measures ANOVA was conducted to investigate potential differences in fixation duration and count between the TBI and non-TBI groups.

#### 3.1.1. Behavioural Data

As demonstrated in Table 2, the descriptive statistics indicated that the group with TBI had lower overall accuracy scores on the EET compared to the group without TBI.

The analysis showed a significant main effect of emotion, (*F* (6, 204) = 6.85, *p* ≤ 0.001, *η_p_*^2^ = 0.17), but the interaction between emotion and group was non-significant (*F* (6, 204) = 1.08, *p* = 0.375, *η_p_*^2^ = 0.03). The test of between-subjects effects was also significant (*F* (1, 34) = 20.28, *p* ≤ 0.001, *η_p_*^2^ = 0.37), with the group means indicating that the TBI participants had significantly fewer correct responses across the EET compared to the non-TBI group. When comparing the descriptive statistics, it appeared that the TBI group scored lower on negative emotions compared to positive, particularly interpreting sad, anxious, and revolted displays of emotion, compared to the non-TBI group, although these differences were not statistically significant.

#### 3.1.2. Eye-Tracking Data


**Fixation duration across the EET in seconds**


The group with TBI had shorter fixation durations to the eyes and nose compared to participants without TBI, while both groups displayed similar fixation durations to the mouth (Table 3).

The analysis showed a significant main effect of emotion (*F* (1.83, 62.17) = 38.99, *p* ≤ 0.001, *η_p_*^2^ = 0.53), and a significant interaction between emotion and AOI (*F* (4.19, 142.35) = 12.29, *p* ≤ 0.001, *η_p_*^2^ = 0.27). The main effect of AOI (*F* (1.70, 57.87) = 0.09, *p* = 0.884, *η_p_*^2^ = 0.003), and the interactions between emotion and group (*F* (1.83, 62.17) = 1.67, *p* = 0.198, *η_p_*^2^ = 0.05), AOI and group (*F* (1.70, 57.87) = 2.17, *p* = 0.130, *η_p_*^2^ = 0.06), and emotion, AOI and group (*F* (4.19, 142.35) = 1.26, *p* = 0.289, *η_p_*^2^ = 0.04) were all non-significant. The tests of between-subjects effects were also not significant (*F* (1, 34) = 2.84, *p* = 0.101, *η_p_*^2^ = 0.08).

The significant interaction between emotion and AOI was explored through three one-way ANOVAs, with emotion as the independent variable (IV), and fixation duration as the dependent variable (DV). There was a significant effect of eyes (*F* (1.40, 48.90) = 36.35, *p* ≤ 0.001, *η_p_*^2^ = 0.51), nose (*F* (2.53, 88.41) = 5.99, *p* = 0.002, *η_p_*^2^ = 0.15) and mouth (*F* (1.40, 48.90) = 36.35, *p* ≤ 0.001, *η_p_*^2^ = 0.51) on fixation duration in response to different facial expressions, with follow-up paired-samples *t* tests in Appendix D. In brief, the post hoc tests indicated that participants had longer fixations on the eyes when viewing sad faces compared to all other emotions. Participants also spent significantly longer looking at the eyes of neutral faces compared to the other emotions (bar sad). The group elicited longer fixations on the nose when viewing angry, sad, and neutral faces, and more fixations on the mouth when looking at happy and sad faces.


**Fixation count across the EET**


The group affected by TBI had fewer fixations compared to those without TBI across all of the emotions, and across most of the AOIs within the emotions (Table 4).

The analysis showed a significant main effect of emotion (*F* (3.40, 115.65) = 41.78, *p* ≤ 0.001, *η_p_*^2^ = 0.55) and a significant interaction between emotion and AOI (*F* (5.85, 198.88) = 10.77, *p* ≤ 0.001, *η_p_*^2^ = 0.24). The main effect of AOI (*F* (1.60, 54.41) = 0.44, *p* = 0.602, *η_p_*^2^ = 0.01) and the interactions between emotion and group (*F* (3.40, 115.65) = 1.79, *p* = 0.146, *η_p_*^2^ = 0.05), AOI and group (*F* (1.60, 54.41) = 2.26, *p* = 0.124, *η_p_*^2^ = 0.06), and emotion, AOI and group (*F* (5.85, 198.88) = 1.88, *p* = 0.088, *η_p_*^2^ = 0.05) were all non-significant. The tests of between-subject effects were also non-significant, (*F* (1, 34) = 2.96, *p* = 0.094, *η_p_*^2^ = 0.08).

The significant interaction between emotion and AOI was explored through three one-way ANOVAs, with emotion as the IV and fixation count as the DV. There was a significant effect of eyes (*F* (1.40, 48.90) = 36.35, *p* ≤ 0.001, *η_p_*^2^ = 0.44), nose (*F* (4.17, 146.02) = 5.87, *p* ≤ 0.001, *η_p_*^2^ = 0.14) and mouth (*F* (3.94, 138.06) = 30.97, *p* ≤ 0.001, *η_p_*^2^ = 0.47) on fixation count in response to different facial expressions, with follow-up paired-samples *t* tests presented in Appendix E. In summary, the group had more fixations on the eyes and nose when viewing neutral and sad faces. The mouth was looked at more when participants viewed happy and sad faces.

#### 3.1.3. Correlations

One-tailed Spearman’s Rho correlations were run to explore the possible relationships between the behavioural data (accuracy for emotion recognition) and eye-tracking data (fixation duration and fixation count). When the groups were combined and the *α*-level had been adjusted to account for multiple comparisons (*α* = 0.01), there were no significant correlations (all, *r*’s ≤ 0.29, all *p*’s ≥ 0.043). When the groups were separated, there were also no significant correlations for the group with TBI (all *r*’*s* ≤ −0.23, *p* ≥ 0.352) or the non-TBI group (all *r*’*s* ≤ 0.58, *p* ≥ 0.013).

### 3.2. Social Inference-Minimal

A 2 (TBI vs. no TBI group) * (3—simple sarcasm, paradoxical sarcasm, sincere) ANOVA was conducted to investigate group differences during the understanding of conversational meanings. A further 2 * (4—intentions, meaning, beliefs, feelings) ANOVA was conducted to investigate the difference between the groups in understanding the different facets of social interactions. With eye-tracking data, a 2 * (3) * (3) mixed-design ANOVA was conducted to investigate potential differences in fixation duration and count between the TBI and control groups.

#### 3.2.1. Behavioural Data


**Accuracy for conversation style**


As demonstrated in Table 5, the descriptive statistics indicated that the TBI group had lower overall accuracy scores across the conversational styles (simple sarcasm, paradoxical sarcasm and sincere) compared to the control group.

There was no significant effect of conversation style (*F* (1.59, 52.47) = 1.37, *p* = 0.260, *η_p_*^2^ = 0.04) and no interaction between conversation style and group (*F* (1.59, 52.47) = 1.63, *p* = 0.203, *η_p_*^2^ = 0.05). The tests of between-subjects effects were significant (*F* (1, 33) = 21.03, *p* ≤ 0.001, *η_p_*^2^ = 0.39), and referring to the estimated marginal means, the group with TBI scored significantly lower across the three conversation styles of the SI-M (*M* = 15.54, *SD* = 3.65) compared to the control group (*M * = 18.55, *SD* = 1.94).


**Accuracy for comprehension probes**


The descriptive statistics for the four different comprehension probes (beliefs, meanings, intentions, feelings) suggested that the group affected by TBI were less accurate in understanding what the actor was trying to do, what they were trying to say, what they were thinking, and what they were feeling (Table 6). See Appendix B for a comprehensive description of the TASIT behavioural metrics.

There was a significant effect of comprehension probe (*F* (2.38, 78.36) = 3.19, *p* = 0.038, *η_p_*^2^ = 0.09), but there was no interaction between comprehension probe and group (*F* (2.38, 78.36) = 1.37, *p* = 0.261, *η_p_*^2^ = 0.04). To further explore the significant effect of comprehension probe, independent *t* tests were conducted. When post hoc correction was applied, the intentions, meaning, beliefs, and feelings probes were significant (*p* ≤ 0.001). The tests of between-subjects effects were significant (*F* (1, 33) = 17.67, *p* ≤ 0.001, *η_p_*^2^ = 0.35), and referring to the estimated marginal means, the group with TBI scored significantly lower across the four comprehension probes of the SI-M (*M * = 11.67, *SD * = 2.29) compared to the control group (*M * = 13.84, *SD * = 1.00).

#### 3.2.2. Eye-Tracking Data


**Fixation duration across the SI-M in seconds**


The mean scores indicated that participants spent more time fixated on the mouth than the nose and eyes during simple sarcasm and sincere videos (Table 7). The sincere videos appeared to elicit the longest fixations, followed by the sarcastic and then paradoxical videos.

The analysis showed a significant main effect of conversational style (*F* (1.34, 44.16) = 23.59, *p* ≤ 0.001, *η_p_*^2^ = 0.42) and AOI (*F* (2, 66) = 6.29, *p* = 0.003, *η_p_*^2^ = 0.16). The interactions between conversation style and group (*F* (1.34, 44.16) = 0.25, *p* = 0.78, *η_p_*^2^ = 0.01), AOI and group (*F* (1.78, 58.82) = 0.39, *p* = 0.657, *η_p_*^2^ = 0.01), conversation style and AOI (*F* (2.39, 78.98) = 2.62, *p* = 0.070, *η_p_*^2^ = 0.07), and conversation style, AOI and group (*F* (2.39, 78.98) = 0.15, *p* = 0.894, *η_p_*^2^ = 0.01) were all non-significant. The tests of between-subjects effects were also not significant (*F* (1, 33) = 0.52, *p* = 0.476, *η_p_*^2^ = 0.02).

To explore the significant main effect of conversation style, two-tailed paired-samples *t* tests were conducted, and the post hoc correction was applied. Participants generated significantly longer fixation durations during the sarcastic videos compared to paradoxical videos (*t* (34) = 5.67, *p* ≤ 0.001, *d* = 0.96, CI = 0.61–1.30), sincere videos compared to sarcastic videos (*t* (34) = −3.00, *p* = 0.005, *d* = 0.51, CI = −1.82–−0.35), and sincere videos compared to paradoxical videos (*t* (34) = −6.50, *p* ≤ 0.001, *d* = 1.09, CI = −2.68–−1.41).

Additional two-tailed paired-samples *t* tests were conducted to investigate the significant main effect of AOI. When the post hoc correction was applied, participants exhibited significantly longer fixations on the mouth (*M * = 2.30, *SD* = 1.69) compared to the eyes (*M * = 1.13, *SD* = 1.07), (*t* (34) = −3.24, *p* = 0.003, *d* = 0.55, CI = −1.90–−0.44).


**Fixation count across the SI-M**


From the mean scores presented in Table 8, there appeared to be a similar pattern for both the TBI and control participants, with both groups having higher fixation counts for the sarcastic and sincere videos compared to paradoxical videos.

The analysis showed a significant main effect of conversational style (*F* (1.39, 45.91) = 22.40, *p* ≤ 0.001, *η_p_*^2^ = 0.40). The other main effect of AOI (*F* (1.69, 55.77) = 2.96, *p* = 0.068, *η_p_*^2^ = 0.08) and the interactions between conversation style and group (*F* (1.39, 45.91) = 0.37, *p* = 0.617, *η_p_*^2^ = 0.01), AOI and group (*F* (1.69, 55.77) = 0.38, *p* = 0.648, *η_p_*^2^ = 0.01), conversation style and AOI (*F* (2.26, 74.71) = 2.40, *p* = 0.091, *η_p_*^2^ = 0.07), and conversation style, AOI and group (*F* (2.26, 74.71) = 0.44, *p* = 0.671, *η_p_*^2^ = 0.01) were all non-significant. The tests of between-subjects effects were also non-significant (*F* (1, 33) = 0.002, *p* = 0.965, *η_p_*^2^ ≤ 0.001).

To explore the significant main effect of conversation style, two-tailed paired-samples *t* tests were conducted, with post hoc corrections applied. Participants generated significantly more fixations during the sarcastic videos compared to paradoxical videos, (*t* (34) = 7.24, *p* ≤ 0.001, *d* = 1.22, CI = 1.83–3.26), and during sincere (*M * = 5.73, *SD* = 3.41) videos compared to paradoxical videos (*t* (34) = −6.30, *p* ≤ 0.001, *d* = 1.06, CI = −4.97–−2.54).

#### 3.2.3. Correlations

One-tailed Spearman’s Rho correlations were used to investigate potential relationships between the behavioural data (accuracy for conversation style and comprehension probe) and eye-tracking data (fixation duration and fixation count). When the groups were combined, and once the α-level had been adjusted to account for multiple comparisons (*α* = 0.01), there were no significant correlations (all *r*’s ≤ 0.001, all *p*’s ≥ 0.110). When the groups were separated, there were still no correlations for the group with TBI (all *r*’*s* ≤ 0.51, *p* ≥ 0.032), but there were significant correlations for the control group between the duration of fixation on the eyes and intention comprehension probes (*r* = 0.74, *p* = 0.001), number of fixations on the eyes and intention comprehension probes (*r* = 0.73, *p* = 0.001), and between the duration of fixation on the eyes and simple sarcasm comprehension probes (*r* = 0.69, *p* = 0.002).

### 3.3. Social Inference-Enriched

A 2 (TBI vs. no TBI group) * (2—sarcastic and lie) ANOVA was conducted to investigate group differences during the understanding of conversational meanings. A further 2 * (4—intentions, meanings, beliefs and feelings) ANOVA was conducted. Again, this analysis investigated the difference between the groups in understanding different facets of social interactions. A 2 * (2) * (3—eyes, nose and mouth AOI) repeated-measures ANOVA was conducted to investigate potential differences in the number and duration of fixations between the TBI and control groups.

#### 3.3.1. Behavioural Data


**Accuracy for Conversational Style**


As demonstrated in Table 9, the mean scores indicated that the group affected by TBI had lower overall accuracy scores across the two conversational styles (sarcastic and lie) compared to the control group.

The effect of conversation style was non-significant (*F* (1.00, 33.00) = 0.64, *p* = 0.428, *η_p_*^2^ = 0.02), but there was a significant interaction between conversation style and group (*F* (1.00, 33.00) = 9.86, *p* = 0.004, *η_p_*^2^ = 0.23). A simple effects analysis revealed that the group with TBI scored significantly lower on the sarcastic vignettes (*M * = 22.56, *SD * = 4.77) compared to controls (*M * = 29.41, *SD * = 3.37) (*p* ≤ *0*.001). The tests of between-subjects effects were also significant (*F* (1, 33) = 16.31, *p* ≤ 0.001, *η_p_*^2^ = 0.33). Referring to the estimated marginal means, the group affected by TBI had significantly lower accuracy scores across the two conversation styles of the SI-E (*M * = 23.94, *SD * = 4.78) compared to the control group (*M * = 28.58, *SD * = 3.24).


**Accuracy for comprehension probes**


The descriptive statistics for the four different comprehension probes (intentions, meanings, beliefs, and feelings) suggested that the group living with TBI were less accurate in understanding what the actor was trying to do, what they were trying to say, what they were thinking, and what they were feeling (Table 10).

There was a significant effect of comprehension probe, (*F* (2.46, 81.19) = 3.25, *p* = 0.034, *η_p_*^2^ = 0.09) and a significant interaction between comprehension probe and group, (*F* (2.46, 81.19) = 4.69, *p* = 0.008, *η_p_*^2^ = 0.12). A simple effects analysis revealed that the group with TBI scored significantly lower on the ‘intention’ probes (*M * = 11.28, *SD * = 2.49) compared to controls (*M * = 14.82, *SD * = 1.63) (*p* ≤ 0.001) and the same for the ‘meaning’ probes (TBI *M * = 11.17, *SD * = 2.60; control group *M * = 14.47, *SD * = 1.70) (*p* ≤ *0*.001). The tests of between-subjects effects were also significant (*F* (1, 33) = 21.41, *p* ≤ 0.001, *η_p_*^2^ = 0.39). Referring to the estimated marginal means, the group affected by TBI scored significantly lower across the four comprehension probes of the SI-E (*M * = 11.97, *SD * = 2.34) compared to the control group (*M * = 14.47, *SD * = 1.63).

#### 3.3.2. Eye-Tracking Data


**Fixation duration across the SI-E in seconds**


Table 11 shows that individuals with TBI had shorter fixations on the eyes, nose and mouth, in terms of duration, compared to controls.

The analysis showed a significant main effect of conversational style, (*F* (1, 33) = 44.18, *p* ≤ 0.001, *η_p_*^2^ = 0.57), with participants exhibiting longer fixations for sarcastic videos compared to lie videos. The main effect of AOI (*F* (1.22, 40.16) = 0.92, *p* = 0.405, *η_p_*^2^ = 0.03) and the interactions between conversation style and group (*F* (1.00, 33.00) = 0.002, *p* = 0.969, *η_p_*^2^ ≤0.01), AOI and group (*F* (1.22, 40.16) = 0.01, *p* = 0.991, *η_p_*^2^ ≤ 0.01), conversation style and AOI (*F* (1.61, 53.26) = 2.54, *p* = 0.099, *η_p_*^2^ = 0.07), and conversation style, AOI and group (*F* (1.61, 53.26) = 1.42, *p* = 0.249, *η_p_*^2^ = 0.04) were all non-significant. The tests of between-subjects effects were also non-significant, (*F* (1, 33) = 1.13, *p* = 0.296, *η_p_*^2^ = 0.03).


**Fixation count across the SI-E**


From the descriptive statistics presented in Table 12, the TBI cohort appeared to produce smaller numbers of fixations on all AOI across the sarcastic and lie conditions compared to control participants. Both the TBI and control participants generated smaller fixation counts during the videos depicting lies compared to videos wherein the actors were being sarcastic.

The analysis showed a significant main effect of conversational style, (*F* (1, 33) = 75.77, *p* ≤ 0.001, *η_p_*^2^ = 0.70), with participants displaying more fixations during the sarcastic vignettes compared to vignettes depicting lies. The main effect of AOI (*F* (1.36, 44.95) = 0.28, *p* = 0.673, *η_p_*^2^ = 0.01) and the interactions between conversation style and group (*F* (1, 33) = 0.08, *p* = 0.780, *η_p_*^2^ = 0.002), AOI and group (*F* (1.36, 44.95) = 0.20, *p* = 0.732, *η_p_*^2^ = 0.01), conversation style and AOI (*F* (1.58, 52.23) = 1.29, *p* = 0.279, *η_p_*^2^ = 0.04), and conversation style, AOI and group (*F* (1.58, 52.23) = 1.49, *p* = 0.236, *η_p_*^2^ = 0.04) were all non-significant. The tests of between-subjects effects were also non-significant (*F* (1, 33) = 0.76, *p* = 0.390, *η_p_*^2^ = 0.02).

#### 3.3.3. Correlations

One-tailed Spearman’s Rho correlations explored the relationships between the behavioural data (accuracy for conversation style and comprehension probe) and eye-tracking data (fixation duration and fixation count). When the groups were combined, and once the α-level had been adjusted to account for multiple comparisons (α = 0.01), there were no significant correlations (*r* ≤ 0.39, *p ≥* 0.022). When the groups were separated, there were still no significant correlations for the group with TBI (all, *r*’s ≤ 0.56, all *p*’s ≥ 0.016) or controls (all, *r*’s ≤ 0.56, all *p*’s ≥ 0.019). Although the analysis did not reach the 0.01 set for post hoc significance, the results are nearing significance, and are significant at the typically accepted α level of 0.05.

## 4. Discussion

We compared the emotion recognition and social inference abilities of participants with TBI to those without TBI. As expected, the TBI group were significantly less accurate in identifying emotions on the EET than those without TBI. In line with our previous work [20], there was a distinct deficit for negative compared to positive emotions, particularly interpreting sad, anxious, and revolted displays of emotion. Individuals with TBI also scored significantly lower across all three of the conversation styles of the SI-M, in comparison to the control group, with simple-sarcasm vignettes scoring the lowest, followed by sincere, and then paradoxical/sarcastic exchanges. This effect was mirrored in the SI-E, where the group with TBI were significantly poorer at recognising sarcastic conversations compared to those without TBI. Interestingly, there was no significant difference between the two groups during the lie vignettes. These findings are consistent with McDonald and colleagues, who reported that participants with TBI displayed impairments in deciphering sarcastic exchanges, but scored in a typical range when interpreting lies [14]. These results imply that while TBI diminishes the ability to comprehend sarcasm, the aptitude to recognise lies remains intact. This model has previously been proposed by McDonald [46], and is based on the conceptual differences between lies and sarcasm. For example, it has been proposed that understanding sarcasm requires more cognitive effort compared to understanding sincere and untruthful exchanges, mainly because understanding sarcasm necessitates first-order theory of mind (ToM; understanding what someone is thinking or feeling) and second-order ToM (predicting what one person thinks or feels about what another person is thinking or feeling) [47]. Analysis of the comprehension probes on both the SI-M and SI-E also demonstrated that the group with TBI had difficulty understanding the actor’s intentions, feelings, beliefs, and the meaning of the conversation, compared to the group without TBI. In summary, these results align with prior research that highlights disruptions in emotion recognition and social inference following a traumatic brain injury (TBI) [11,14,15,16,39,48]. However, our study contributes to the field by pioneering an investigation into the potential involvement of visual impairments in this issue, utilising ecologically valid assessments.

There were no significant differences between the groups in terms of eye fixation patterns, and no associations between fixation patterns and behavioural accuracy scores. This differs from previous studies that have reported TBI groups exhibiting abnormal fixation patterns compared to non-TBI groups [23,24,25]. Interestingly, these studies utilised static stimuli that have been criticised for potential low ecological validity, for instance, lack of motion, depth cues, and contextual cues [26,49,50,51,52]. Evidence has demonstrated that static and dynamic stimuli activate different brain networks [53,54], as well as different levels of attention. For example, Kujawa and colleagues reported that individuals with impaired consciousness resulting from brain injury exhibited higher visual attention to dynamic compared to static stimuli [55]. Others have also shown that eye fixation patterns differ in normative populations in response to dynamic and static stimuli [56,57]. It appears that when more real-world dynamic social stimuli are implemented, there are no group dissimilarities in terms of eye fixation patterns between those with and without TBI, a finding we replicated in a previous study [20]. This article provides insights into the implications of the absence of deficits in eye-tracking when considering the processing of emotional and social information following TBI. Our study suggests that individuals with TBI may not encounter challenges in perceiving emotional and social stimuli, rather they encounter difficulties in processing this information effectively. This observation aligns with the existing body of research indicating that individuals with TBI often exhibit a multitude of cognitive and socioemotional deficits. For instance, extant research has reported a substantial correlation between theory of mind, a socioemotional skill, and performance on phonemic fluency in a TBI sample but not in a control group [58]. Phonemic fluency is a measure of executive functioning that places specific demands on cognitive flexibility and self-regulation. These findings are consistent with other research, which suggests that deficits in certain aspects of executive functioning may partially contribute to impairments in social cognition in TBI populations as well as in other populations wherein impairment in social functioning is a central feature, such as autism spectrum disorder [59,60].

These results might suggest that the challenges in everyday emotion recognition and social inference following a TBI are not necessarily linked to visual alterations. Nevertheless, in the present study, we did observe distinctions between our TBI and non-TBI groups at a descriptive level. Additionally, considering previous reports on abnormal eye fixation patterns post-TBI in response to static stimuli, it may be rash to entirely dismiss the possibility that visual changes might exert some influence on social cognition. It seems that there is a multifaceted relationship between vision and social cognition after TBI, necessitating further investigation, potentially with larger sample sizes and increased statistical power to detect subtle group differences. Currently, it remains unclear what is driving these social impairments, but speculatively, they may be attributed to a combination of low-level visual functions and higher-order functions such as executive function, working memory load [48], education and/or executive functioning [39].

There was a significant interaction between AOI and emotion during the EET, with visual strategies appearing to reflect attention to the most diagnostic area of the face for each emotion (e.g., the mouth for identifying happiness, and the nose for recognising disgusted faces). These findings indicate a goal- and stimulus-driven influence on eye gaze patterns, as distinct patterns of fixation count and duration were observed for both emotional and neutral faces and for conversational styles. This finding is in line with previous research [30,31,32]. During the SI-M, all participants exhibited longer fixations and higher fixation counts during the sarcastic videos compared to paradoxical and sincere compared to paradoxical videos. For the group without TBI, there was a correlation between the duration of the fixation on the eyes and intentions, the number of fixations on the eyes and intentions, and between the duration of the fixation on the eyes and simple sarcasm. Recognising sincere, sarcastic, and deceptive communication requires the integration of verbal, non-verbal, and paralinguistic cues, and is paramount to everyday interactions. Despite this, previous research exploring this complex area has heavily relied on written and static stimuli. Our findings conflict with existing research, which reported that individuals had longer-lasting fixations on nonliteral compared with literal interactions [61]. Gaining a deeper understanding of typical literal and non-literal language perception can inform existing theoretical frameworks as well as aid clinical rehabilitation areas, such as TBI, where these skills are frequently diminished [62].

The primary limitation of the present research was the relatively small sample size. Future work might include larger samples, with more female participants, or more conservative alpha levels during significance testing. Demographically, TBI and non-TBI groups were matched for age, sex, and years of education, but those with TBI fell below the non-TBI group on IQ measures. This is typical for TBI populations [63], but if social impairments post-TBI are related to working memory deficits, then future studies could control for this. Another area of future research could be to explore whether disruption to frontal or high-order brain areas could uniquely contribute to social cognition deficits post-TBI.

## 5. Conclusions

Our results demonstrate two things. First, eye fixation patterns for dynamic social stimuli do not statistically differ between individuals with and without TBI, but do differ on a descriptive level; second, dynamic emotion recognition and social inference abilities are impaired after TBI. These combined findings seem to indicate that impairments are not related to abhorrent eye fixation patterns, and must be driven by a disruption outside of the low-level visual system. Our research also highlights the important distinction between static and dynamic stimuli when trying to map structure to function in eye-tracking research, and we urge other research teams to implement dynamic stimuli in the future.

## Figures and Tables

**Table 1 behavsci-13-00816-t001:** Mean, standard deviations (SD), significance (*p*), and effect size (Cohen’s *d*) for demographic variables for TBI and non-TBI groups.

Demographic Variable	TBI Group Mean (SD)	Non-TBI Group Mean (SD)	*p*	*d.*
Gender	m = 15, f = 3	m = 15, f = 3		
Age at test	44.94 (11.69)	43.83 (12.26)	0.696	0.09
Age at injury	36.44 (13.79)			
Post-injury years	8.50 (8.68)			
Years of education	14.83 (4.25)	5.56 (3.65)	0.389	0.18
Verbal IQ	84.06 (18.71)	95.33 (8.66)	0.007 *	0.77
Performance IQ	91.00 (17.50)	104.72 (11.64)	0.150	0.94
Full IQ score	90.25 (19.69)	100.06 (10.44)	0.025 *	0.65

Note: * indicates a significant *p*-value at 0.05.

**Table 2 behavsci-13-00816-t002:** Descriptive statistics for the number of correct responses for the TBI and non-TBI groups for the EET.

TASIT EET Score	TBI Mean (SD)	Non-TBI Mean (SD)
Overall Correct	19.67 (3.99)	24.28 (1.60)
Happy	3.11 (1.13)	3.28 (0.67)
Surprised	3.22 (0.81)	3.67 (0.49)
Neutral	2.00 (0.91)	2.72 (0.83)
Sad	2.72 (1.32)	3.50 (0.71)
Angry	3.00 (1.08)	3.56 (0.51)
Anxious	2.89 (1.32)	3.94 (0.24)
Revolted	2.72 (1.02)	3.56 (0.62)

**Table 3 behavsci-13-00816-t003:** Descriptive statistics for the TBI and non-TBI group for fixation duration across the EET in seconds.

EET Emotions	Groups Combined Mean (SD)	TBI Mean (SD)	Non-TBI Mean (SD)
**Angry**	3.97 (3.01)	2.92 (2.12)	5.03 (3.43)
Eyes	1.15 (1.25)	0.92 (1.08)	1.37 (1.40)
Nose	1.59 (1.80)	0.93 (1.16)	2.26 (2.10)
Mouth	1.23 (0.90)	1.07 (0.90)	1.40 (0.90)
**Revolted**	3.76 (3.06)	3.01 (1.99)	4.50 (3.76)
Eyes	0.78 (0.97)	0.72 (0.97)	0.85 (0.99)
Nose	1.43 (2.05)	0.82 (1.04)	2.05 (2.60)
Mouth	1.54 (1.56)	1.48 (1.35)	1.61 (1.78)
**Anxious**	2.44 (2.74)	1.53 (2.10)	3.35 (3.05)
Eyes	0.41 (0.61)	0.51 (0.74)	0.32 (0.44)
Nose	1.43 (1.99)	0.60 (1.06)	2.27 (2.36)
Mouth	0.60 (0.87)	0.42 (0.76)	0.77 (0.96)
**Happy**	4.41 (2.35)	4.04 (2.32)	4.79 (2.38)
Eyes	0.93 (1.13)	0.62 (0.96)	1.24 (1.23)
Nose	0.86 (1.02)	0.69 (0.68)	1.02 (1.28)
Mouth	2.62 (1.70)	2.73 (1.74)	2.53 (1.71)
**Sad**	8.56 (6.68)	6.92 (5.48)	10.21 (7.50)
Eyes	3.84 (3.53)	3.40 (3.91)	4.28 (3.16)
Nose	2.16 (3.16)	1.05 (1.13)	2.37 (3.99)
Mouth	2.56 (2.60)	2.47 (2.59)	2.65 (2.67)
**Surprised**	2.18 (1.95)	1.64 (1.77)	2.72 (2.01)
Eyes	0.63 (1.05)	0.70 (1.28)	0.55 (0.79)
Nose	0.94 (1.22)	0.51 (0.66)	1.38 (1.49)
Mouth	0.61 (0.68)	0.43 (0.46)	0.79 (0.82)
**Neutral**	4.84 (2.92)	4.36 (2.76)	5.32 (3.08)
Eyes	1.92 (2.22)	1.85 (2.36)	1.98 (2.14)
Nose	1.59 (1.49)	1.13 (1.23)	2.06 (1.61)
Mouth	1.33 (1.23)	1.38 (1.51)	1.27 (0.90)
**Overall (emotions combined)**	3.82 (2.82)	2.93 (2.32)	4.69 (3.05)
Eyes	0.98 (1.21)	0.87 (1.29)	1.57 (1.58)
Nose	1.37 (1.65)	0.72 (0.89)	1.57 (1.56)
Mouth	1.47 (1.16)	1.35 (1.16)	1.68 (1.88)

**Table 4 behavsci-13-00816-t004:** Descriptive statistics for the TBI and non-TBI group for fixation count across the EET.

EET Emotions	Groups Combined Mean (SD)	TBI Mean (SD)	Non-TBI Mean (SD)
**Angry**	8.80 (5.52)	6.54 (3.75)	11.06 (6.15)
Eyes	2.67 (2.76)	1.93 (2.14)	3.42 (3.16)
Nose	2.98 (2.74)	1.69 (1.70)	4.26 (3.01)
Mouth	3.14 (1.98)	2.92 (1.99)	3.38 (1.99)
**Revolted**	7.12 (4.93)	6.36 (4.07)	7.89 (5.68)
Eyes	1.68 (1.66)	1.66 (1.68)	1.70 (1.69)
Nose	2.75 (3.00)	2.02 (2.35)	3.48 (3.45)
Mouth	2.69 (2.13)	2.68 (2.02)	2.71 (2.29)
**Anxious**	4.21 (4.25)	3.00 (3.51)	5.41 (4.68)
Eyes	1.02 (1.24)	1.12 (1.39)	0.91 (1.11)
Nose	2.02 (2.52)	0.92 (1.13)	3.11 (3.05)
Mouth	1.17 (1.65)	0.96 (1.65)	1.39 (1.65)
**Happy**	9.22 (4.19)	8.79 (3.99)	9.64 (4.45)
Eyes	2.17 (2.12)	1.42 (1.38)	2.92 (2.49)
Nose	2.25 (1.77)	2.07 (1.65)	2.43 (1.92)
Mouth	4.79 (2.60)	5.30 (2.88)	4.29 (2.26)
**Sad**	12.81 (8.24)	11.15 (7.00)	14.47 (9.21)
Eyes	5.22 (3.77)	4.96 (4.08)	5.49 (3.53)
Nose	3.44 (3.98)	2.21 (2.51)	4.67 (4.81)
Mouth	4.15 (3.26)	3.99 (3.09)	4.32 (3.49)
**Surprised**	4.29 (3.28)	3.37 (2.89)	5.21 (3.47)
Eyes	1.08 (1.16)	1.02 (1.19)	1.13 (1.16)
Nose	1.87 (2.15)	1.19 (1.32)	2.54 (2.60)
Mouth	1.34 (1.29)	1.16 (1.15)	1.53 (1.42)
**Neutral**	9.56 (5.42)	7.81 (4.34)	11.31 (5.93)
Eyes	3.75 (3.72)	2.98 (3.08)	4.52 (4.21)
Nose	3.03 (2.46)	2.12 (2.10)	3.94 (2.50)
Mouth	2.78 (2.03)	2.71 (2.29)	2.84 (1.78)
**Overall**	7.38 (4.61)	5.82 (3.72)	8.94 (4.97)
Eyes	2.02 (1.84)	1.57 (1.58)	2.48 (2.01)
Nose	2.54 (2.44)	1.57 (1.56)	3.51 (2.80)
Mouth	2.82 (1.80)	2.68 (1.88)	2.96 (1.76)

**Table 5 behavsci-13-00816-t005:** Descriptive statistics for the accuracy of the TBI and control groups for the different conversational constructs during the SI-M.

SI-M Score	TBI Mean (SD)	Control Mean (SD)
Simple sarcasm	15.28 (3.63)	17.71 (2.11)
Paradoxical sarcasm	15.94 (4.01)	18.18 (2.24)
Sincere	15.39 (3.31)	19.76 (1.48)

**Table 6 behavsci-13-00816-t006:** Descriptive statistics for the accuracy scores of the TBI and non-TBI groups for the four comprehension probes during the SI-M.

SI-M Score	TBI Mean (SD)	Non-TBI Mean (SD)
Intentions	11.39 (2.73)	13.82 (1.07)
Meaning	11.78 (2.21)	13.83 (0.95)
Beliefs	11.11 (2.47)	13.71 (1.05)
Feelings	12.89 (1.75)	14.00 (0.94)

**Table 7 behavsci-13-00816-t007:** Fixation duration for SI-M across the conversation styles.

Conversational Style	Overall Mean (SD)	TBI Mean (SD)	Control Mean (SD)
**Simple sarcasm**	1.70 (1.54)	1.77 (1.47)	1.63 (1.65)
Eyes	0.34 (0.54)	0.44 (0.65)	0.23 (0.37)
Nose	0.55 (0.67	0.51 (0.56)	0.58 (0.78)
Mouth	0.81 (0.79)	0.82 (0.75)	0.81 (0.86)
**Paradoxical sarcasm**	0.74 (0.66)	0.80 (0.62)	0.68 (0.72)
Eyes	0.21 (0.26)	0.26 (0.30)	0.15 (0.19)
Nose	0.29 (0.39)	0.24 (0.30)	0.35 (0.48)
Mouth	0.24 (0.28)	0.30 (0.33)	0.18 (0.20)
**Sincere**	2.79 (1.82)	3.02 (2.24)	2.53 (1.26)
Eyes	0.59 (0.82)	0.68 (1.00)	0.49 (0.59)
Nose	0.95 (0.92)	1.01 (0.98)	0.89 (0.88)
Mouth	1.25 (1.23)	1.34 (1.51)	1.15 (0.88)

**Table 8 behavsci-13-00816-t008:** Descriptive statistics for the AOI for the TBI and control groups across the three conversational styles (fixation count).

Conversational Style	Overall Mean (SD)	TBI Mean (SD)	Control Mean (SD)
**Sarcasm**	4.52 (3.21)	4.62 (2.74)	4.41 (3.72)
Eyes	0.95 (1.27)	1.22 (1.49)	0.67 (0.96)
Nose	1.55 (1.46)	1.48 (1.76)	1.61 (1.75)
Mouth	2.02 (1.58)	1.92 (1.43)	2.13 (1.76)
**Paradoxical sarcasm**	1.97 (1.48)	2.12 (1.44)	1.82 (1.56)
Eyes	0.62 (0.63)	0.76 (0.71)	0.46 (0.52)
Nose	0.75 (0.78)	0.73 (0.69)	0.76 (0.89)
Mouth	0.61 (0.50)	0.63 (0.56)	0.59 (0.46)
**Sincere**	5.73 (3.41)	5.44 (3.71)	6.04 (3.16)
Eyes	1.67 (2.06)	1.55 (1.62)	1.79 (2.49)
Nose	1.94 (1.56)	1.91 (1.67)	1.98 (1.49)
Mouth	2.12 (1.69)	1.98 (1.90)	2.27 (1.48)

**Table 9 behavsci-13-00816-t009:** Descriptive statistics for the behavioural data of the TBI and control groups during the sarcastic and lie conditions of the SI-E.

SI-E Score	TBI Mean (SD)	Control Mean (SD)
Sarcastic	22.56 (4.77)	29.41 (3.37)
Lie	25.33 (4.38)	27.76 (3.11)

**Table 10 behavsci-13-00816-t010:** Descriptive statistics for the behavioural data of the TBI and control groups for the four comprehension probes during the SI-E.

SI-E Score	TBI Mean (SD)	Control Mean (SD)
Intentions	11.28 (2.49)	14.82 (1.63)
Meaning	11.17 (2.60)	14.47 (1.70)
Beliefs	13.33 (1.81)	14.47 (0.72)
Feelings	12.11 (2.47)	14.12 (2.45)

**Table 11 behavsci-13-00816-t011:** Descriptive statistics for the duration of fixations on the AOI for the TBI and control groups across the two conversational styles of the SI-E.

Conversational Style	Overall Mean (SD)	TBI Mean (SD)	Control Mean (SD)
**Sarcasm**	0.98 (0.92)	0.85 (0.70)	1.11 (1.10)
Eyes	0.37 (0.45)	0.31 (0.37)	0.43 (0.53)
Nose	0.35 (0.39)	0.29 (0.30)	0.41 (0.47)
Mouth	0.26 (0.25)	0.25 (0.19)	0.28 (0.30)
**Lie**	0.38 (0.57)	0.25 (0.30)	0.51 (0.75)
Eyes	0.14 (0.24)	0.12 (0.19)	0.17 (0.30)
Nose	0.11 (0.20)	0.07 (0.11)	0.14 (0.25)
Mouth	0.13 (0.22)	0.06 (0.08)	0.20 (0.29)

**Table 12 behavsci-13-00816-t012:** Descriptive statistics for fixation counts to the AOI for the TBI and control groups across the two conversational styles of the SI-E.

Conversational Style	Overall Mean (SD)	TBI Mean (SD)	Control Mean (SD)
**Sarcasm**	3.90 (2.74)	3.62 (2.04)	4.20 (3.37)
Eyes	1.33 (1.25)	1.21 (1.10)	1.45 (1.42)
Nose	1.39 (1.22)	1.24 (0.83)	1.55 (1.55)
Mouth	1.18 (0.82)	1.16 (0.72)	1.20 (0.95)
**Lie**	1.48 (1.94)	1.12 (0.98)	1.86 (2.58)
Eyes	0.47 (0.63)	0.44 (0.53)	0.49 (0.74)
Nose	0.49 (0.74)	0.35 (0.39)	0.65 (0.98)
Mouth	0.52 (0.77)	0.33 (0.31)	0.72 (1.03)

## Data Availability

Not applicable.

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
