# Peer review of "Dynamic Emotion Recognition and Social Inference Ability in Traumatic Brain Injury: An Eye-Tracking Comparison Study"

_behavsci, 2023, doi:10.3390/bs13100816_

Round 1

Reviewer 1 Report

Brief Summary

The authors test emotion recognition and social interference scores in TBI and matched controls. Importantly, authors use dynamic, as opposed to static cues to test emotional and social impairments. Although people with TBI were worse at accurately identifying emotions and worse at distinguishing social cues, there were no differences in eye-tracking between groups. Thus, the authors conclude that low-level visual strategies do not appear to be contributing to higher-level emotional and social perception deficits seen post-TBI.

General Comments

Overall, the article contains clearly interpretable findings that add new information to the traumatic brain injury field. The paper could be improved by including a more thoughtful discussion on what the lack of eye-tracking differences might mean about emotional/ social processing and by including figures to illustrate the results in the table.  

1.    In the discussion can you expand on what the lack of eye-tracking deficits might mean in regard to how emotional/social information is processed. Does this suggest that people with TBI do not have a problem sensing emotional/ social stimuli, but do have a problem processing this information? Could this show a relationship with cognitive deficits described in this population?

2.    The discussion also warrants more of a focus on brain area impacted by TBI. Could disruption to frontal or high-order brain areas be uniquely contributing to this effect? Could brain area have been controlled for in the statistical analysis (i.e., covariate) and would this change your results?

3.    Could figures be added to help illustrate the data provided in each table? This would greatly help the reader to visualize the significant differences between groups.

Specific Comments

66: The third point does not make sense to the reader yet. This aim is too specific compared to the prior aims.

81: hemorrhage

100: Exclusion criteria: How many participants were excluded based on these criteria?

109: What does None mean? This was an exclusion criterion, so none were included, but were any excluded for the above reasons?

128: participants

131: study; the eyes, nose, and mouth

147: Why were only correct scores included? Couldnt incorrect scores have interesting findings too?

150: Table 1 could significant scores be indicated with an asterisk or by bold font?

182: Approximately how long did a session take?

183: Be more specific about which kinds of tests were conducted for each section below (i.e. independent samples t-test).

Table 2: Add statistic test (parametric or nonparametric test) and identify significant differences.

215: Formatting of headers change- keep consistent in this section.

Table 7: Sincere category numbers are shifted to the left.

Author Response

Dear reviewer, 

We would like to thank you for taking the time to read our manuscript and for your helpful and insightful comments. We believe that your feedback has enhanced the clarity of the manuscript. 

1. In the discussion can you expand on what the lack of eye-tracking deficits might mean in regard to how emotional/social information is processed. Does this suggest that people with TBI do not have a problem sensing emotional/ social stimuli, but do have a problem processing this information? Could this show a relationship with cognitive deficits described in this population? We have expanded on this point in the discussion.

2. The discussion also warrants more of a focus on brain area impacted by TBI. Could disruption to frontal or high-order brain areas be uniquely contributing to this effect? Could brain area have been controlled for in the statistical analysis (i.e., covariate) and would this change your results? This is an interesting area but as we did not explore this in our study, we feel that any discussion would be speculative and would not add anything concrete to the discussion. As our groups are quite small (12 predominately frontal lobe pathology and 6 pathology which does not encroach on either frontal lobes, or occipital cortex, but may be present in other cortical or subcortical brain regions) we feel we would not have sufficient power to detect effects. We have added a sentence in the discussion as a possibility for future research.

3. Could figures be added to help illustrate the data provided in each table? This would greatly help the reader to visualize the significant differences between groups. Given that the article is already lengthy at 25 pages, and considering that including figures would be duplicating the data already provided in the tables, we believe it is preferable to adhere to using only tables.

Specific comments

66: The third point does not make sense to the reader yet. This aim is too specific compared to the prior aims. We agree and we have amended the aim so it is more general. 

81: hemorrhage. Amended. 

100: Exclusion criteria: How many participants were excluded based on these criteria? The clinicians did not keep track of this detail and we have included this information in the manuscript now. 

109: What does “None” mean? This was an exclusion criterion, so none were included, but were any excluded for the above reasons? We have clarified this in the manuscript. 

128: participants’. Amended. 

131: study; the eyes, nose, and mouth… Amended. 

147: Why were only correct scores included? Couldn’t incorrect scores have interesting findings too? We agree that incorrect scores could have provided interesting findings. However, as our research objective in this emotion recognition study revolved around assessing the accuracy and effectiveness of emotion recognition processes, by focusing on correct responses, we are directly addressing this objective and providing insights into the participants' abilities to recognize emotions accurately. Additionally, limiting the analysis to correct responses allows for a fair and meaningful comparison between groups. It ensures that any observed differences or trends in emotion recognition performance and associated eye tracking are more likely to reflect genuine variations in their abilities rather than chance or other confounding factors.

150: Table 1 – could significant scores be indicated with an asterisk or by bold font? We have added an asterisk to indicate significant scores. 

182: Approximately how long did a session take? We have added this detail. 

183: Be more specific about which kinds of tests were conducted for each section below (i.e. independent samples t-test). We feel this point has been addressed.

Table 2: Add statistic test (parametric or nonparametric test) and identify significant differences. Table 2 outlines descriptive statistics. We think this might be in reference to Table 1 so we have added the detail of the test underneath. 

215: Formatting of headers change- keep consistent in this section. We think this might be in reference to 3.1 and 3.1.1. We have used the MDPI template predetermined heading formats and believe they adhere to the MDPI formatting requirements. However, we did notice that there were omissions in section 3 which we have amended.

Table 7: Sincere category numbers are shifted to the left. Apologies, this was a formatting issue and has now been rectified. 

Author Response

Dear Reviewer,

We extend our gratitude for dedicating your valuable time to review our manuscript and for sharing your insightful and constructive comments. Your feedback has undeniably improved the overall clarity of the manuscript.

1) It would be good to present the experimental procedure in one place, in the form of a diagram or at least in bullet points. We have added a diagram (Appendix C) to illustrate the experimental process.

2) The version of Tobii Studio software used and the main parameters of the computer on which it was installed should be added. The name and the version of the statistical software used in the research should also be provided. We have added the details that we have to the Method section. Unfortunately, the eye-tracking equipment and PC that were used to run the study have been destroyed and so we have added the information that we are aware of. We have added the version of SPSS we used.

3) In lines 128-130, the sentence should be corrected because the eye tracker recorded participants' eye movement whereas metrics are calculated by Tobii Studio software. We have amended this sentence for clarity.

4) The numbering of subsections in Chapter 3 is missing. We have amended this.

5) In line 415 there is an incorrect number in the table. This has been amended.